# Smartphone Addiction among University Students in Light of the COVID-19 Pandemic: Prevalence, Relationship to Academic Procrastination, Quality of Life, Gender and Educational Stage

**DOI:** 10.3390/ijerph191610439

**Published:** 2022-08-22

**Authors:** Ismael Salamah Albursan, Mohammad Farhan Al. Qudah, Hafidha Sulaiman Al-Barashdi, Salaheldin Farah Bakhiet, Eqbal Darandari, Sumayyah S. Al-Asqah, Heba Ibraheem Hammad, Mohammed M. Al-Khadher, Saleem Qara, Sultan Howedey Al-Mutairy, Huthaifa I. Albursan

**Affiliations:** 1Department of Psychology, College of Education, King Saud University, Riyadh 11451, Saudi Arabia; 2The Research Council, Sultanate of Oman, Muscat 123, Oman; 3Department of Special Education, College of Education, King Saud University, Riyadh 11451, Saudi Arabia; 4Department of Psychology, College of Education, Qassim University, Qassim 52571, Saudi Arabia; 5Department of Psychology, Princess Alia College, AL Balqa Applied University, Amman 11821, Jordan; 6Department of Basic Sciences, College of Arts and Sciences, Middle East University, Amman 11831, Jordan; 7Department of Educational Technology, College of Education, King Saud University, Riyadh 11451, Saudi Arabia; 8Faculty of Medicine, Hashmite University, Zarka 13133, Jordan

**Keywords:** smartphone addiction, COVID-19, prevalence rates

## Abstract

The current study aims to identify the level and proportions of smartphone addiction, and academic procrastination among university students in the light of the Corona pandemic; identify the differences in smartphone addiction, academic procrastination, and quality of life according to gender and stage of study; and revealing the predictive ability of academic procrastination and quality of life for smartphone addiction. **Methods**: 556 male and female students from Saudi universities participated in the study, whose ages ranged from 18 to 52 years. Measures of academic procrastination and quality of life were used, in addition to the Italian scale of smartphone addiction, which was translated and checked for validity and reliability. **Results**: The results revealed that 37.4% of the sample were addicted to smartphone use, while 7.7% had a high level of procrastination, and 62.8% had an average level of procrastination. The results did not show statistically significant differences in smartphone addiction and quality of life according to gender and educational stage, while there were statistically significant differences in academic procrastination according to gender in favor of males, and according to stage of education in favor of undergraduate students. The results also revealed a statistically significant positive relationship between academic procrastination and smartphone addiction, and a statistically significant negative relationship between smartphone addiction and quality of life. A negative relationship between quality of life and academic procrastination was found. The results also revealed that addiction to smartphones could be predicted through academic procrastination and quality of life.

## 1. Introduction

The sudden outbreak of Coronavirus Disease 2019 (COVID-19) has had a dramatic effect on the mental health of the public. Smartphone users under coronavirus lockdown reported significant increases in the time they are spending on their devices. The number of smartphone users worldwide today surpasses three billion and is forecast to further grow by several hundred million in the next few years [1]. Although individuals expect several benefits from using smartphones, such as virtual social interaction, entertainment, and access to information [2,3] their interaction via smartphones has increased and, in some cases, has hampered their day-to-day activities [2]. Smartphone addiction is similar to internet addiction and can be considered a behavioral addiction [4,5]. Smartphone addiction was introduced to our lives as a consequence of the interaction between people and mobile information and communication technologies [6].

Home quarantine and social distancing during the Corona pandemic have led to an increase in internet usage globally. Some studies correlate COVID-19 related anxiety and depression with an increase in smartphone addiction cases. Elhai, et al. [7] correlated COVID-19 anxiety with severity of problematic smartphone use, depression, and anxiety. Using established cut-off scores, 12% of participants were identified with at least moderate depression and 24% with moderate anxiety. Using structural equation modelling, COVID-19 anxiety related to problematic smartphone use severity, mediating relations between general anxiety and problematic smartphone use severity. However, controlling problematic smartphone use for general anxiety and depression severity, COVID-19 anxiety no longer predicted problematic smartphone use severity. According to [8] psychoactive substances and other reinforcing behaviors (e.g., gambling, video gaming, watching pornography) are often used to reduce stress and anxiety and/or to alleviate the depressed mood. The tendency to use such substances and engage in such behaviors more extremely as putative coping strategies in crises like the COVID-19 pandemic is considerable. Duan, et al. [9] demonstrated the psychological effects on children and adolescents associated with the epidemic. Nine factors were associated with increased levels of depression, such as smartphone addiction, internet addiction, and residence in Hubei province.

In fact, social media played a central role in individuals’ social lives, including to get updated information about COVID-19 pandemic, to maintain their communications, and sometimes to distract themselves from the fear and anxiety created by the pandemic [10]. Gudiño, et al. [11] evaluated the use of social media before and during the COVID-19 lockdown using a Spanish sample. The results showed that during the lockdown, there was an increase in the number of hours spent per week using social media, especially Facebook, WhatsApp and YouTube, to share images, videos and audio messages. Likewise, [12] reported that users used social media as the primary medium for information and there was a shift towards digital medium in Cyprus. Similar findings were obtained in European countries and China, where excessive internet use was observed during the lockdown. This increase was especially pronounced in the case of smartphones, followed by tablets and PCs.

Studies across different cultures showed that COVID-19 pandemic has psychological, social, and economic effects. For example, [13] and [14] confirmed the negative psychological effects of COVID-19 in Turkey and India. Baloglu, et al. [10] investigated the psychological, social, somatic, and economic effects of COVID-19 using a heterogeneous sample, and reported differences based on gender, educational attainment, socioeconomic statuses, and geographical regions of Turkey. People are socially impacted by this pandemic due to the inability to visit their peers, colleagues, and distant family members. Besides economic impact of the lockdown was much severe. Besides, social isolation and lockdowns have led to emotional and mental health issues such as stress, depression, fear, anxiety, insomnia, and emotional exhaustion [14].

The relationships between excessive smartphone use and poor psychological well-being, anxiety, and feelings of loneliness have been reported among university students in international studies [15]. Individuals developing behavioral addiction to smartphones, usually neglect other tasks and duties and have lower quality of life, which points to the importance to investigate the problematic smartphone use among university students and if it could be predicted by academic procrastination and quality of life.

### 1.1. Smartphone Addiction Prevalence Rates, Predictors and Negative Impacts among University Students

Smartphone addiction can be defined as “the inability to control the smartphone use despite negative effects on users” [5]. Some studies exploring the prevalence of smartphone addiction among university students have found high percentages [16,17,18,19,20,21]. Buctot, et al. [17] found that 62.6% of participants had a smartphone addiction, while [21] found that 14.3% of participants had a smartphone addiction. Another study looking at signs of mobile phone addiction among university students in Belarus and Poland concluded that about 10.4% of Belarusian students and about 22.9% of Polish students showed symptoms of smartphone addiction [19]. Kwak, et al. [20] also reported that about 70% of adolescents had moderate to severe addiction to smartphones. Chen, et al. [18] found the prevalence of smartphone addiction among medical college students was 29.8%. AlBarashdi [16] found that 33.1% of university students in Oman had a smartphone addiction. Aljomaa, et al. [22] revealed that smartphone addiction percentage among university students was 48%. Albursan, et al. [23] found that the prevalence of smartphone addiction among Jordanian students was 59.8% followed by Saudi students 27.2%, then Sudanese students 17.3%, and Yemeni students 8.6%. The overall prevalence in the four countries was 27.7%. Al-Qudah, et al. [24] revealed that the frequency of smartphone addiction among participants was 33.2%.

Researchers revealed that smartphone addiction could be predicted; people who suffer from psychological and emotional problems such as loneliness, depression, isolation and distraction easily become addicted to technological devices such as smartphones [4,25]. In addition, negative relationships with family members and peers are major factors in smartphone addiction [20]. Parental neglect or negative parental relationship can directly lead to smartphone addiction [20,26], or it can cause depression which results in smartphone addiction [20,27] as a way to escape real life. Moreover, maladjustment and being unable to cope with social situations can lead people to spend most of their time on smart devices as a mean to avoid social conflict [20], therefore becoming addicted to smartphones and their applications.

Studies also found that excessive use of smartphones by university students had negative academic, psychological, and physical impacts [28,29]. Individuals addicted to smartphones show psychological symptoms such as compulsive behavior, tolerance, withdrawal, and anxiety [3,28,30]. It is also noticeable that students addicted to smartphones exhibit lack of attention, lack of self-control, hyperactivity, and anger [29]. Excessive smartphone use can also have negative physical impacts. A decrease in physical activity and movement can lead to obesity among addicts [29]. Other symptoms include carpal tunnel syndrome, backaches, migraine headaches, dry eyes, loss of sound sleep, and thumb strain injuries [25]. These symptoms can appear in all smartphone addicts but can be worse in university students.

### 1.2. Smartphone Addiction Relationship to Academic Procrastination

Academic procrastination is the voluntary delay of action on academic tasks despite expecting to be worse off for that delay. It is so prevalent that, according to some estimates, 50–80% of college students procrastinate moderately or severely [31]. Ebadi & Shakoorzadeh [32] found that over half of students nearly always or always procrastinate. In addition, results showed that males and females procrastinate at the same rate

Previous research has explored whether smartphone addiction could positively predict academic procrastination [33,34,35,36,37]. Akinci [33] reported that smartphone addiction is a significant positive predictor for academic procrastination. Students with problematic smartphone use were also expected to neglect and delay academic responsibilities. Al-Qudah, et al. [34] found three levels of academic procrastination among Saudi university students: the highest percentage of procrastination was for the moderate level 83.6%, followed by the high level 9.7% of procrastination, while the lowest percentage of procrastination was for the low level 6.7%. According to [37], smartphone addiction negatively affects students’ physical and mental health, with consequences such as interpersonal communication problems and academic failure. Usage time partially mediated the relationship between smartphone addiction and procrastination.

The proportions of explanation for indirect effects were 20.32% and 24.70%, respectively. Smartphone addiction mediated the relationships between self-regulation and both academic anxiety and academic procrastination [38]. Wang, et al. [39] revealed that procrastination partially mediated the relation between sensation seeking and adolescent smartphone addiction Esichaiku, et al. [40] showed that academic procrastination and smartphone addiction were predictable through academic stress. Furthermore, some studies have also demonstrated that smartphone addiction leads to mental health problems, for instance, depression, stress, anxiety and poor sleep quality could accompany the emergence of addiction behaviors [41,42].

### 1.3. Smartphone Addiction Relationship to Quality of Life

The concept of quality of life broadly encompasses how an individual measure the “goodness” of multiple aspects of their life [43]. These evaluations include one’s emotional reactions to life occurrences, disposition, sense of life fulfilment and satisfaction, and satisfaction with work and personal relationships [44].

Recent studies have demonstrated a negative correlation between smartphone addiction and quality of life among university students [17,45,46,47,48,49,50,51,52,53]. Safa & Majeed [52] found that smartphone addiction and loneliness showed a significant negative relationship with quality of life in late adolescents and early adults. Furthermore, according to [51], smartphone addiction is increasing among university students, which can affect both emotional intelligence and self-regulation and in turn their quality of life. Likewise, Gao, et al. [46] found that smartphone addiction and depression both significantly affected neuroticism and quality of life. The direct effect of neuroticism on quality of life was significant, and the chain-mediating effect of smartphone addiction and depression was significant. However, in a recent study, Gao, et al. [47] found that quality of life played a partial mediator role in the relationship between parent-child relationship and smartphone use disorder. University students who were addicted to smartphone use had significantly lower scores across all quality-of-life domains [50]. A significant inverse relationship exists between smartphone addiction scores and the quality-of-life scores for physical, mental, and social aspects. Smartphone addiction scores are significantly higher for females, bachelors, and married students than other groups of students. The smartphone addiction score determines 6% of the variance in quality of life. Alongside addiction, smartphone overuse may negatively influence the physical, mental, and social aspects of students’ quality of life [53]. In addition, Buctot, et al. [17] found a significant negative correlation between smartphone addiction and health-related quality of life as well as its subdomains: physical well-being, psychological well-being, and school environment, but not with autonomy and parent or peer social support. Kumcagiz [49] suggested that quality of life negatively correlated with smartphone addiction in Turkish high school students. Likewise, Demir & Sumer, et al. [45] found a significant negative correlation between smartphone addiction and quality of life in migraine patients. In addition, Citó, et al. [54] reported that smartphone addiction was a predictor of quality of life, before and during COVID-19 lockdown, and affected the student academic performance.

### 1.4. The Relationship between Smartphone Addiction, Academic Procrastination, and Quality of Life by Gender and Education Stage

Gender plays a moderating role in the influence of smartphone addiction on academic procrastination and quality of life. Liu, et al. [37] found that addictive behavior related to procrastination was more profound in male groups than in female groups. Buctot, et al. [17] found that smartphone addiction was more prevalent among Filipino male students. They spent more hours on smartphones on weekdays and weekends, and as a result, their quality of life was lower. Factors associated with smartphone addiction in male students were the use of game apps, anxiety, and poor sleep quality. Significant factors for female undergraduates were the use of multimedia applications, the use of social networking services, depression, anxiety, and poor sleep quality. Nayak [55] indicated that although female students are more likely to use smartphones, the effect of their use on academic achievement appears to be more pronounced among male students. Female students were found to show hardly any effect of smartphone addiction, unlike the male students who were found to neglect work, feel anxious, and lose control of themselves. Lee & Kim [21] found no significant differences in smartphone addiction between gender groups. In Gulf Cooperation Council (GCC) countries, Aljomaa, et al. [22], and Al-Qudah, et al. [24] revealed a significant gender difference in smartphone addiction in favor of males. However, Albursan, et al. [23] found that females displayed greater smartphone addiction than males.

Few studies have explored the differences between undergraduate and postgraduate students in smartphone addiction, academic procrastination, and quality of life. Masthi [56] found that Facebook dependency was more commonly observed among postgraduate students. Zhang, et al. [57] found that mobile phone dependence of university students in first-year significantly predicted poor mental health status in their third year. Moreover, college adjustment at Year 2 significantly mediated the effect of mobile phone dependence in Year 1 on mental health status in third year. Ickes, et al. [58] found no significant differences between undergraduate and graduate students in academic stress levels. However, social support as a coping strategy was the most important variable explaining the differences between undergraduate and graduate students. Aljomaa, et al. [22] revealed a significant gender difference in smartphone addiction in favor of undergraduate students.

The researchers noted that several studies were conducted in different countries to explore the effect of smartphone addiction on academic procrastination or quality of life. However, there is a lack of studies regarding the relationship among the three variables together. More specifically, no study as far as the researchers know, examined the effect of both academic procrastination and quality of life on smartphone addiction, and the magnitude of that effect in light of the COVID-19 pandemic, particularly for university students. This could be another angle of viewing this issue. Furthermore, demographic variables such as gender and educational stage factors were not considered as mediating factors for the relationship among these variables in previous studies, and it is important to explore this association.

### 1.5. The Current Study

Although smartphone addiction, academic procrastination, and quality of life have attracted the attention of academics [39,59,60,61], few studies have focused on investigating the prevalence of smartphone addiction among university students, its relationship to academic procrastination and its impact on their quality of life. Similarly, few studies have investigated whether demographic variables can predict smartphone addiction, academic procrastination, and quality of life. A few studies were conducted to investigate association between academic procrastination and quality of life and smartphone addiction in the Arabic environment, and the different cultural contexts could add to explaining this association.

The significance of the present study stems from the fact that it is one of the rare studies that specifically addressed the problem of smartphone addiction and its association with both academic procrastination and quality of life, among university students, considering moderating factors of gender and educational level. Furthermore, examining the effects of academic procrastination and quality of life on smartphone addiction during a pandemic is expected to help management of such problems during similar crisis. Countries do not face the pandemic at the same level, so one country’s experience can help others.

Therefore, this study aims to determine the prevalence of smartphone addiction and academic procrastination, and quality of life levels among university students, and to investigate the relationship between these three variables and some demographic variables. In line with these aims, the research questions are as follows:

Q1: What are the prevalence rates of smartphone addiction and academic procrastination among university students in light of the COVID-19 pandemic?

Q2: Are there differences in smartphone addiction, academic procrastination, and quality of life based on gender?

Q3: Are there differences in smartphone addiction, academic procrastination, and quality of life by stage of education (undergraduate and graduate students)?

Q4: Is it possible to predict smartphone addiction of university students through quality of life and academic procrastination?

## 2. Materials and Methods

### 2.1. Participants

A total of 556 students from Saudi universities participated in this study (130 students from King Saud University = 23.38%; 114 students from Umm Al-Qura University = 20.50%; 110 students from King Khalid University = 19.78%, 103 students from the University of Hail =18.50%, and 99 students from Dammam University = 17.80%), of whom 190 were males (34.2%), and 366 were females (65.8%), aged between 18 and 55 years (M = 31.36, SD = 9.69). The sample was distributed over the academic stages: 342 Bachelor’s degree students (61.5%), and 214 Postgraduate students (39.5%).

### 2.2. Data Collection Procedure

The researchers prepared an electronic questionnaire that included the three measures of the study in addition to a personal information form that contained the demographic variables. To gather data, this questionnaire (that included these measures) was sent through social media (WhatsApp and Twitter) during April 2020 to the students of the four universities from different regions in Saudi Arabia. The researchers used the snowball sampling method, where the questionnaire was disseminated to several students, from different educational stages, in these universities asking them to share the link with their peers. Students agreed to participate voluntarily as a convenience sampling (non-probabilistic) technique that was employed. The questionnaire included information about the purpose of the study and the voluntary and anonymous nature of the students’ participation. Participants were told that the data collected would be confidential and used only for scientific research, and that they were not required to give their names. The electronic system required completion of all the items, and no data were missing. The administration of the study measures coincided with the presence of strict health measures in the Kingdom of Saudi Arabia that called for social distancing and staying at home except for specific times and needs. At that time, travel was prevented between cities, gatherings were forbidden, and distance learning was used due to closure of schools and universities, as well as remote working, travel and flight restrictions.

### 2.3. Instruments

A personal information form and three different data collection tools were used in the research, which are: Smartphone Addiction Inventory (SPAI), Academic Procrastination Scale (APS), and Quality of Life Scale. The personal information form was developed by the researchers to collect data on participants’ demographic variables of gender, stage, age, and university. The data collection instruments used in this study were developed by different researchers and were suitable, reliable and valid instruments according to the aims of the study. Their details are as follows:

#### The Smartphone Addiction Inventory (SPAI)

The SPAI was created by [62] to measure smartphone addiction. It has been translated into other languages and it displayed good reliability and validity across different countries and languages. Lin, et al. [62] conducted SPAI on a sample consisted of *283* male and female students aged between 20.96 and 24.96 years. The results of exploratory factor analysis (EFA) identified four dimensions, which are compulsive behavior, functional impairment, abstinence, and tolerance. Internal consistency indices of SPAI were all satisfactory for the overall score and its factors (Cronbach’s alpha = 0.94, 0.87, 0.88, 0.81, and 0.72), respectively. The four subscales had moderate to high correlations (0.56–0.78). Simo’-Sanz, et al. [63] conducted the Spanish version of the SPAI on a sample of 2958 university participants aged 18 years and above, and reported good model fit and reliability for the scale and its four factors (Cronbach’s alpha = 0.95, 0.86, 0.89, 0.86, and 0.71), respectively.

However, the Italian version of the SPAI, which is a 24 item self-report questionnaire designed to assess smartphone addiction in adolescence, was conducted by [64] on 485 university students. Confirmatory factor analysis (CFA) showed better fit for the five-factor model (time spent on smartphone, compulsivity to use smartphone, daily life interference, craving for smartphone use, and sleep interference), that explained 53% of the total variance. Two items were removed from the inventory, because they had low factor loadings. Good reliability has been found for the total score and each of its corresponding factors (Cronbach’s alpha ranged between 0.70 to 0.81). Wang, et al. [39] conducted a Chinese version of SPAI on a sample of Chinese adolescents, and the goodness of fit indicators supported the five-factor model, and it yielded good internal consistencies (all Cronbach’s alphas > 0.70). On the other side, Khoury, et al. [65] conducted the Brazilian version of the SPAI on a sample of university students, and CFA results confirmed the one-factor model with good fit indices.

In this study, the English version of the SPAI, revised by [64], was used after getting permission from authors. The SPAI was translated into Arabic by two bilingual researchers, and the authenticity of the translation was confirmed by back- translation from Arabic into English by another researcher. The original English version and the re-translated version showed only minor adjustments that were made in translation to suit the Arab environment. It consisted of 24 items and utilized a four-point Likert-type: strongly agree (4), agree (3), do not agree (2), strongly disagree (1). Scores range was from 24 to 96, and researchers used the midpoint method in scaling to identify addiction; a score of 60 or above was considered to show addiction, that is a mean per item of 2.5 or higher. The validity of the scale was verified by the corrected correlation coefficients of items with the total score, which ranged between 0.65 and 0.80. EFA, through principal component analysis, was conducted, and multiple criteria were used to determine the number of inventory factors; including eigenvalues greater than 1, percentage of variance explained by the factors, and factor loadings [66]. The results showed that there was only one general factor that had eigen value greater than 1. The percentage of the explained variance of the first factor was 53.64, which represents 90% of the total variance before rotation, and was 31.30, which represents approximately 53% of the total variance 59.72, after rotation. This indicates that the scale is one-dimensional and not multi-dimensional. In addition, the internal consistency of the Arabic version of SPAI was high (Cronbach’s alpha = 0.96).

### 2.4. Academic Procrastination Scale

The researchers used the Academic Procrastination Scale (APS) of [34] that consisted of (21) items, which is an edited version of [67], that was based on other well-known scales (such as: [68,69]. The APS has good psychometric properties. Ref. [67] reported that correlation coefficients between the items and the scale scores were acceptable and ranged between (0.36–0.73), and the internal consistency of the total scale was high (Cronbach’s alpha = 0.90). In addition, Al-Qudah, et al. [34] administered the modified Arabic APS on a sample of 50 university students and reported good psychometric properties for it, and sowed that correlation coefficients between the items and the total score ranged between (0.39–0.77), and were statistically significant (*p* < 0.05); test-retest reliability coefficient for the total scale was high (0.91), and internal consistency was good (Cronbach’s alpha = 0.85).

The current study confirmed these results for the modified APS [34], where the researchers conducted it on a sample of 100 university students (other than the study sample). All Participants answered each item according to 5-point Likert type scale in which responses ranged from very high (5), high (4), moderate (3), low (2) to very low (1); and total scores ranged between 21–105. The score means were extracted and categorized to determine students’ grades on the APS as follows: 2.38 or less = low procrastination; from 3.59 to 2.37 = medium procrastination; from 3.58 and above = high procrastination. The correlation coefficients between the items and the scale score were acceptable and ranged between (0.33–0.81), and the internal consistency of the total scale was high (Cronbach’s alpha = 0.90). These results confirmed good validity and reliability for the Arabic version of APS, that was used in this study

#### Quality of Life Scale

The World Health Organization (WHO) quality of life instrument, the WHOQOL, consists of 100 items and captures many subjective aspects of quality of life [70]. This study used the abbreviated version of it (WHOQOL-BREF) which is one of the best-known instruments that has been developed for cross-cultural comparisons of quality of life and is available in more than 40 languages [71]. The WHOQOL-BREF is a 26-item instrument. It contains two items on overall quality of public life and public health, and 24 items with one item from each of the 24 facets from the WHOQOL-100. The 26 items produce 4 domains: physical health, mental health, social relations, and physical environment, in addition to the overall quality of life and health satisfaction facet. 

This study used the Arabic version of WHOQOL-BREF of [72] who translated the instrument and adapted it into Arabic, and reported its psychometric properties, where the results showed that the scale had four factors that explained 76.57% of the total variance, and the test reliability was high (test-retest reliability coefficient = 0.89, and Cronbach’s Alpha = 0.93). Likert scale of 5 points was used to estimate severity and frequency and to assess the characteristics chosen, with the following alternatives: very good (5), somewhat good (4), neither bad nor good (3), somewhat bad (2), very bad (1); and very satisfied (5), somewhat satisfied (4), neither satisfied nor dissatisfied (3), somewhat dissatisfied (2), never satisfied (1). Scores as a whole ranged between 26 and 130, where a higher score on the scale was an indication of higher quality of life. 

### 2.5. Data Analysis

The data was coded and entered using IBM SPSS Statistics 25 software. To answer the research questions, frequency, percentage, mean and standard deviations, T test and Pearson correlation coefficients were calculated to examine the relationships between the study variables. Finally, multi-regression analysis was used to identify the predictive power of academic procrastination and quality of life of smartphone addiction.

## 3. Results

### 3.1. Academic Procrastination and Smartphone Addiction Rates for the Individual Study Sample

Table 1 shows the averages and rates of academic procrastination and smartphone addiction among university students. It appears from Table 1 that the percentage of smartphone addiction is more than a third of the sample (37.4%). The percentage of those with high academic procrastination was low (7.7%), while the largest proportion showed moderate procrastination (62.8%).

### 3.2. The Differences between Males and Females in Addiction to Smartphones, Academic Procrastination and Quality of Life

A *t*-test for independent samples was used to detect the results of male-female differences in smartphone addiction, academic procrastination, and quality of life, and Table 2 illustrates this.

There was no statistically significant difference at the 0.05 level between males and females (*p* = 0.638 > 0.05) in addiction to smartphones. In academic procrastination, there were statistically significant differences between males and females in favor of males (*p* = 0.002 < 0.05), but with a low impact size (0.28), according to [73]. As for quality of life, there was no statistically significant differences between male and female students (*p* = 0.339 > 0.05).

### 3.3. Differences between Undergraduate and Graduate Students in Smartphone Addiction, Academic Procrastination and Quality of Life

The T test for independent samples was used to calculate the differences between undergraduate and graduate students in smartphone addiction, academic procrastination and quality of life, and Table 3 shows the values of T and their significance by education stage for each of the three variables.

It is clear from Table 3 that there are no statistically significant differences at the 0.05 level between undergraduate and graduate students in smartphone addiction (*p* = 0.464 > 0.05). In academic procrastination, there were statistically significant differences in favor of undergraduate students (*p* = 0.038 < 0.05), but the size of the impact is low (0.180), according to [73]. As for quality of life, there was no statistically significant differences between undergraduate and postgraduate students (*p* = 0.383 > 0.05).

### 3.4. Predicting Addiction to Smartphones among University Students through Academic Procrastination and Quality of Life

Pearson correlation coefficients between study variables were calculated. The results showed that there are statistically significant negative relationships at the 0.01 level between the variables of quality of life and smartphone addiction (r = −0.39), and statistically significant positive relationship between smartphone addiction and academic procrastination (r = 0.44) at the 0.01 level. The relationship between the variables of academic procrastination and quality of life was negative (r = −0.46) and statistically significant at the 0.01 level.

To reveal the predictive power of the variables of academic procrastination and quality of life for smartphone addiction, a progressive multiple regression analysis was used, and Table 4 shows the results of the analysis.

Collinearity issues were confirmed with Variance Inflation Factor (VIF); values were below 10 (Average VIF = 1.27), which indicates that there is no multicollinearity problem. Table 4 shows the results of multiple regression using the stepwise method to enter the variables, and the results showed that addiction to smartphones can be predicted from academic procrastination and quality of life (R2 = 0.20 for step 1, F (1.55) = 135.29, *p* < 0.001; for step 2, ΔR2 = 0.04, F (2.55) = 87.67, *p* < 0.01).

Academic procrastination was the best predictor of addiction to smartphones, as it was able to explain 0.20 of addiction variance, while in the second model consisting of academic procrastination and quality of life, they were jointly able to explain 0.24 of the variance of smartphone addiction. Thus, the quality-of-life variable has been able to predict an additional 0.04 of the variance, which is a statistically significant value at the 0.01 level. This prediction can be expressed by the following equation:

Smartphone addiction = 52.70 + 0.40 × (academic procrastination) − 0.25 × (quality of life)

## 4. Discussion

First, our findings revealed a high prevalence rate of smartphone addiction among our sample (37.4%). Compared to previous studies, this percentage is close to what [16] found (33.1%). Yet, it is more than what other researchers found such as [18] 29.8%, Karjewska-Kulak, et al. [19] 22.9%, and [21] 14.3%. However, it is less than what other studies found, such as [17], who reported that 62.6% of the participants had smartphone addiction. Likewise, Kwak, et al. [20] reported that about 70% of adolescents had moderate to severe addiction to smartphones. Furthermore, our findings also showed that the percentage of those with high academic procrastination was 7.7%, while the percentage of those with moderate procrastination was 62.8%. Compared to some of the previous studies, this percentage is close to the levels found by [31,32], but less than [34], who found that the percentage of procrastination among Saudi university students was for the moderate level, which was 83.6%.

Second, our findings revealed no statistically significant differences between male and female students in smartphone addiction and quality of life; however, there were statistically significant differences between males and females in academic procrastination, in favor of males. These results are consistent with the findings of other studies such as [37], which found that addictive behavior related to procrastination was more profound in male groups than in female groups. Other studies also found that smartphone addiction was more prevalent among male students (e.g., [17,21,22,23,24]. However, our results are partially consistent with the findings of other studies such as [55], which indicated that although female students are more likely to use smartphones, the effect of the use on academic achievement appears to be more pronounced among male students.

Third, our findings revealed that there were no statistically significant differences between undergraduate and postgraduate students in smartphone addiction and quality of life, while there were statistically significant differences between undergraduate and postgraduate students in academic procrastination, in favor of undergraduate students. These results are inconsistent with the findings revealed by some previous studies ([22,57]), which reported significant differences in smartphone addiction, in favor of undergraduate students; and with the findings revealed by [56] study, which showed that Facebook dependency was more commonly observed among postgraduate students.

Fourth, our results showed that there were statistically significant negative relationships between quality of life and smartphone addiction, and between quality of life and academic procrastination, while there was statistically significant positive relationship between smartphone addiction and academic procrastination. In addition, our results revealed that addiction to smartphones could be predicted from academic procrastination and quality of life. These results are consistent with the findings of previous studies that demonstrated a negative correlation between smartphone addiction and quality of life among university students [17,45,46,48,49,50,51,52,53]. These results are also consistent with the findings of studies that found that smartphone addiction could positively predict academic procrastination (e.g., [33,35,36,37] and quality of life (e.g., [54]).

## 5. Conclusions

Our findings revealed that the behaviors of addiction to smartphones and academic procrastination were both major problems facing respondents. The results of the study revealed that academic procrastination was more prevalent among male students, and undergraduate students were more affected by procrastination than postgraduate students. Among the most important findings of our study was the active role of the variables of academic procrastination and quality of life in smartphone addiction. Our results also revealed a negative relationship between procrastination and quality of life, a negative relationship between smartphone addiction and quality of life, and a positive relationship between smartphone addiction and academic procrastination. Furthermore, our results showed that addiction to smartphones could be predicted from academic procrastination and quality of life.

## 6. Implications and Limitations

Our findings heightened the need to prepare and design preventive, therapeutic and counselling programs to reduce problematic use of the smartphone and its negative effects, including increased academic procrastination and lower quality of life. Our results also demonstrated the need for more studies to determine the full extent of smartphone addiction and academic procrastination in light of this pandemic, and to identify its negative consequences. This could aid in the design of advisory and preventive programs by social workers and professionals for the harmful effects of smartphone addiction on quality of life. In general, the results of this study may provoke future studies looking at the relationship between smartphone addiction and procrastination by addressing quality of life as an intermediate variable. These could be conducted on similar samples and different age groups at the regional and international levels, and further research related to prevention and intervention programs that address limiting the excessive use of smartphones in light of the pandemic could also be conducted. While other studies confirmed the predictive power of smartphone addiction of academic procrastination and quality of life, our study results confirmed the predictive power of academic procrastination and quality of life of smartphone addiction, and considered differences across gender and educational level groups. Therfore, more advanced predictive models are needed to investigate the causal effect of each of the study variables considering direct, indirect and intermediate effects. Other factors related to contextual differences among countries could also be considered.

Although the results of this study support the relationship between study variables, the study sample was limited, non-random, and taken from students of universities in the Kingdom of Saudi Arabia aged between 18 and 52, which limits the ability to generalize results to different age groups and societies in different cultural contexts. In addition, this study is limited by the data collection means that happened through an electronic survey via social media platforms, WhatsApp and Twitter, because of the Corona pandemic restrictions. Moreover, it is limited to the scales used in the study, which are Smartphone Addiction Inventory (SPAI), Academic Procrastination Scale (APS), and Quality of Life Scale (WHOQOL-BREF). Despite the good psychometric properties for these scales, responses relied on self-report, which may be affected by the tendency among respondents for social desirability. The research also relied on descriptive, comparative and correlative research design from among the cross-sectional studies, which calls for more experimental research in this field using quantitative and qualitative tools, and their application to larger and more varied random samples and different age groups.

## 
Institutional Review Board Statement


The study was conducted according to the guidelines of the Declaration of Helsinki, and approved by the Ethics Committee of Jordan Academicians League (Protocol code 021 and date of approval 10 February2020).

## Figures and Tables

**Table 1 ijerph-19-10439-t001:** The averages and rates of addiction to smartphones and academic procrastination among university students.

Variable	Mean *	SD	Addicted %	Not Addicted	Low AP %	Medium AP %	High AP %
Smartphone addiction	55.31 *	16.11	208	348			
			(37.40%)	(62.60%)			
Academic procrastination	57.56 **	13.56		164		349	43
				(29.50%)		(62.80%)	(7.70%)

* Total score = 96; ** Total score = 105.

**Table 2 ijerph-19-10439-t002:** The differences between males and females in smartphone addiction, academic procrastination, and quality of life.

Variables	Sex	N	Mean	SD	T-Value	Sig.	Effect Size
Smartphone addiction	Male	190	54.87	16.631	0.47	0.638	-
	female	366	55.55	15.85			
Academic procrastination	Male	190	59.91	12.28	3.1	0.002	0.28
	female	366	56.33	14.04			
Quality of life	Male	190	84.29	15.02	0.96	0.339	-
	female	366	82.96	15.87			

**Table 3 ijerph-19-10439-t003:** The differences between undergraduate and graduate students in smartphone addiction, academic procrastination, and quality of life.

Variables	Educational Stage	N	Mean	SD	T-Value	Sig.	Effect Size
Smartphone addiction	Undergraduate	342	55.71	15.58	0.73	0.464	-
	Graduate	214	54.68	16.95			
Academic procrastination	Undergraduate	342	58.5	13.53	2.08	0.038	0.18
	Graduate	214	56.05	13.5			
Quality of life	Undergraduate	342	83.87	16.11	0.87	0.383	-
	Graduate	214	82.68	14.71			

**Table 4 ijerph-19-10439-t004:** Predictive Power of Academic Procrastination and Quality of Life with Addiction to Smartphone.

Variable	B	Std. Error	β	T	Sig.
Step 1					
Constant	25.02	2.68		9.35	<0.001
Academic procrastination	0.53	0.05	0.44	11.63	<0.001
Step 2					
constant	52.7	5.56		9.54	<0.001
Academic procrastination	0.4	0.05	0.33	8	<0.001
Quality of life	−0.25	0.04	−0.24	5.7	<0.001

R^2^ = 0.20 for step 1 (*p* < 0.001); R^2^ = 0 0.24 for step 2 (*p* < 0.01).

## Data Availability

Data will be provided if requested.

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
