# Peer review of "Smartphone Addiction among University Students in Light of the COVID-19 Pandemic: Prevalence, Relationship to Academic Procrastination, Quality of Life, Gender and Educational Stage"

_ijerph, 2022, doi:10.3390/ijerph191610439_

Round 1
Reviewer 1 Report
The article describes an interesting reserach in which the smartphone addiction among University Students in light of the Corona Pandemic was explored. Aims were to (i) determine the prevalence of smartphone addiction and academic procrastination, (ii) analyze the differences in smartphone addiction, academic procrastination and quality of life based on gender, (iii) explore the differences in smartphone addiction, academic procrastination and quality of life by stage of education (undergraduate and graduate students) and (iv) investigate the relationship between these three variables and some demographic variables.
However, there are a number of problems that recommend this article be revised before it should be considered for publication:
- The review of the literature mentions a number of relevant studies, but the large scope and breath of research needs to be indicated even if it is impossible to cite larger numbers of studies. It will be important to note that cultural contexts may determine the results of this research, but it necessary to include works based on population of other countries. For example, the following ones: 1) Social Media and the Pandemic: Consumption Habits of the Spanish Population before and during the COVID-19 Lockdown, 2) Psychological and socio-economic effects of the COVID-19 Pandemic on Turkish Population, 3) The changes in the effects of social media use of Cypriots due to COVID-19 pandemic.
- The title is too long.
- The abstract and the aims of the article are clear, but it is necessary to explain the gap which generated this study. Why is such a description useful or necessary? What does such an analysis provide for readers of the journal? Will this be relevant to a broad, international audience? What else could this analysis teach us?
- The description of the tools used to carry out the research is clear. However, I feel ambiguous about the methodology. On one hand, did the participants know the research purpose? Can't this be a strange variable? On the other hand, in the “participants” section there is some information related to the data collection procedure. It is important to create a new section with this denomination (data collection procedure) to explain a bit more about it.
- The references which appear on the text are not well ordered in the parentheses; for example: Yildiz-Durak, 2017; Sreeniva & Philip, 2019.
- There are some ideas which are repetitive; for example, in this sentence it would be possitive to remove the references from the end because then the authors explained them one by one: “Home quarantine and social distancing during the Corona pandemic have led to an 59 increase in internet usage globally. Some studies correlate COVID-19-related anxiety and 60 depression with an increase in smartphone addiction cases (Duan et al., 2020; Elhai et al., 61 2020; Király et al., 2020)”.
- The beginning of the “participants” section starts with a number (line 232), the authors could use “a total of…”.
- Check typographical errors; 1) the use of “:” after “participants” (line 231), 2) the description of the instruments that appear in bold, 3) the use of data number with 2 or 3 decimals, it is necessary to use the same number of decimal in all data. The use of “.” and “,” in data number.
I hope that these comments, oriented toward formative feedback, will help the authors to improve the text. Good job.
Author Response
Response to Reviewers
Dear Reviewer 1,
Thank you very much for providing us with the supportive and helpful reviews! We will describe in detail our changes and answers.
Point 1: The title is too long.
No changes were made
(Line 1-4)
Response: We respect the reviewer recommendation; however, it is very difficult to do. The title seems to be long, but it should be so, because the study has several variables and questions that need to be addressed in the title. The title of study includes the main dependent variable “Smartphone Addiction” and the other two independent variables “Academic Procrastination, Quality of Life”. Besides, it includes “gender” and “Educational Stage” variables that were addressed in the study, as well as prevalence of Smartphone Addiction” and its relationship to other variables addressed in the study.
Point 2: Explain the gap which generated this study. Why is such a description useful or necessary? What does such an analysis provide for readers of the journal? Will this be relevant to a broad, international audience? What else could this analysis teach us?
Response: Recommended changes were conducted.
(Line 229-237; 245-256)
The gap generated this study was explained, its importance, and how the journal reader and international audience could benefit from it.
Point 3: The review of the literature must cite larger numbers of studies with cultural contexts that may determine the results of this research, but it necessary to include works based on population of other countries.:
1) Social Media and the Pandemic: Consumption Habits of the Spanish Population before and during the COVID-19 Lockdown,
2) Psychological and socio-economic effects of the COVID-19 Pandemic on Turkish Population,
3) The changes in the effects of social media use of Cypriots due to COVID-19 pandemic.
Response: Recommended changes were conducted
(Line 76-102)
A larger number of studies with different cultural contexts from population of other countries were added including all the ones indicated by the reviewer.
Point 4: Did the participants know the research purpose?
-create a new section with this denomination (data collection procedure) to explain a bit more about it.
Response: Recommended changes were conducted
(Line 279-297)
The permission was stated, and a new section “data collection procedure” was added to explain the procedures.
Point 5: There are some ideas which are repetitive; for example, in this sentence it would be positive to remove the references from the end because then the authors explained them one by one: “Home quarantine and social distancing during the Corona pandemic have led to an 59 increase in internet usage globally. Some studies correlate COVID-19-related anxiety and 60 depression with an increase in smartphone addiction cases (Duan et al., 2020; Elhai et al., 61 2020; Király et al., 2020)”.
Response: Recommended changes were conducted
Repetitions were removed
(Line 54-70).
Point 6: Check typographical errors;
1- the use of “:” after “participants” (line 231),
2- the description of the instruments that appear in bold,
3- the use of data number with 2 or 3 decimals, it is necessary to use the same number of decimals in all data.
4-The use of “.” and “,” in data number.
- The beginning of the “participants” section starts with a number (line 232), the authors could use “a total of…”.
Response: Recommended changes were conducted (Line 271; 272; 298; 299-391; 398-460).
All typographical errors were corrected.
Point 7: The references which appear on the text are not well ordered in the parentheses; for example: Yildiz-Durak, 2017; Sreeniva & Philip, 2019.
Response: Recommended changes were conducted, (Line 54; 126).
References in the text were re-ordered.
Reviewer 2 Report
First of all, congratulations to the authors for the organization of the article presented and the theme.
The article talks about the relationship between smartphone addition and academic proscrastination, quality of lige, gender and educational stage, during the corona pandemic.
The article is well organized and easy to read.
The first comment on the article is about the title: Why use corona pandemic and not COVID-19 as described in the keywords?
Next, I would like to make some comments about the scales used:
Regarding the SPAI scale, as written in this article, it is not clear who the original author of the scale is, which version was used for arabic translation and the exploratory factor analysis process.
In fact Lin et al. is the author of SPAI and his article does not identify 5 factors, but 4 factors: “compulsive behavior”, “functional impairment”, “withdrawal” and “tolerance”, they had 283 participants and Cronbach's alpha was 0.94 (only Cronbach's alpha agrees with what is written by the authors). Pavia et al. did the translation of the SPAI into Italian, the evaluation of the psychometric properties and the confirmatory factor analysis (for the Italian population). The Italian version of the SPAI has 5 factors (as described in this article), 485 participants participated in this study (as mentioned in this article) and Cronbach’s alpha of 0.90. So, the way this article is written, it is confusing to understand which version the authors used. Has authorization been requested from the authors for the translation into arabic? It is also not clear the exploratory factor analysis process, which only revealed 1 factor. Clarification is suggested.
Regarding the Academic Procrastination Scale, it is not clear who is the original author of the scale and the psychometric properties of the original scale, nor is it clear who did the translation into arabic (Al-Qudah et al. is not in the references) and what are the psychometric properties of the arabic version. Clarification is suggested.
Regarding the WHO Quality of Life Scale, it is not clear which scale was used (since the WHO has more than one scale; given that the scale used in this study has 26 items, I believe that the WHOQOL-Bref was used. Clarification is suggested.
The following comment refer to the predictive model used. Why use academic procrastination and quality of life to predict smartphone addiction? Doesn't it make more sense for smartphone addiction to predict academic procrastination and quality of life? In fact, this is one of the implications identified by the authors, right in the first sentence of the implications and limitations.
Author Response
Dear Reviewer 2,
Once again, many thanks for the helpful recommendations. Please find our answers and explanations in the following
Point 1: title: Why use corona pandemic and not COVID-19 as described in the keywords?
Response: Recommended changes were conducted. (Line 3)
Point 2: Regarding the SPAI scale, who the original author of the scale is, which version was used for arabic translation
Lin et al. is the author of SPAI and his article does not identify 5 factors, but 4 factors: “compulsive behavior”, “functional impairment”, “withdrawal” and “tolerance”, they had 283 participants and Cronbach's alpha was 0.94.
Pavia et al. did the translation of the SPAI into Italian. The Italian version of the SPAI has 5 factors, 485 participants participated and Cronbach’s alpha of 0.90.
Response: Recommended changes were conducted. (Line 307-384)
Regarding the SPAI scale, the original author of the scale and the version translated to Arabic was clarified. Information regarding the samples and reliability and validity evidence of the original scale, including the factors, were corrected.
The psychometric properties information of Pavia et al. scale were corrected.
Point 3: Has authorization been requested from the authors of SPAI scale for the translation into arabic?
Response: Recommended changes were conducted. (Line 330-331)
The permission was indicated to.
Point 4: Clarify exploratory factor analysis process which only revealed 1 factor for SPAI scale
Response: Recommended changes were conducted. (Line 340-345)
Exploratory factor analysis process for SPAI scale was explained
Point 5: Who is the original author of Academic Procrastination Scale and the psychometric properties of the original scale, who did the translation into Arabic (Al-Qudah et al. is not in the references) and what are the psychometric properties of the Arabic version.
Response: Recommended changes were conducted. (Line 350-371)
The original author of Academic Procrastination Scale and the psychometric properties of the original scale and Arabic scale were clarified. The person who did the translation was clarified too.
Point 6: Which scale was used of the WHO Quality of Life Scale, (I believe that the WHOQOL-Bref with 26 items was used.)
Response: Recommended changes were conducted. (Line 373-385)
The original and brief scales were clarified with more detailes regarding what was used in the current study
Point 7: predictive model used. Why use academic procrastination and quality of life to predict smartphone addiction? Doesn't it make more sense for smartphone addiction to predict academic procrastination and quality of life? In fact, this is one of the implications identified by the authors, right in the first sentence of the implications and limitations.
Response: Recommended changes were conducted. (Line 232-235; 229-238; 535-542)
Mode explanations were provided regarding predictive model used in the study and how it is different from other studies, and why it was used.